# Transcriptomic Changes in Endothelial Cells Triggered by Na,K-ATPase Inhibition: A Search for Upstream Na^+^_i_/K^+^_i_ Sensitive Genes

**DOI:** 10.3390/ijms21217992

**Published:** 2020-10-27

**Authors:** Elizaveta A. Klimanova, Svetlana V. Sidorenko, Polina A. Abramicheva, Artem M. Tverskoi, Sergei N. Orlov, Olga D. Lopina

**Affiliations:** 1Faculty of Biology, Lomonosov Moscow State University, 119234 Moscow, Russia; sidorenko.svetlana.v@yandex.ru (S.V.S.); abramicheva.polina@gmail.com (P.A.A.); tverskoiam@gmail.com (A.M.T.); sergeinorlov@yandex.ru (S.N.O.); od_lopina@mail.ru (O.D.L.); 2Engelhardt Institute of Molecular Biology, Russian Academy of Sciences (RAS), 119991 Moscow, Russia

**Keywords:** endothelial cells, Na, K-ATPase, cardiotonic steroids, [Na^+^]_i_/[K^+^]_i_-ratio, FOS, transcription, G-quadruplexes

## Abstract

Stimulus-dependent elevation of intracellular Ca^2+^ affects gene expression via well-documented calmodulin-mediated signaling pathways. Recently, we found that the addition of extra- and intracellular Ca^2+^ chelators increased, rather than decreased, the number of genes expressed, and that this is affected by the elevation of [Na^+^]_i_/[K^+^]_i_-ratio. This assumes the existence of a novel Na^+^_i_/K^+^_i_-mediated Ca^2+^_i_-independent mechanism of excitation-transcription coupling. To identify upstream Na^+^_i_/K^+^_i_-sensitive genes, we examined the kinetics of transcriptomic changes in human umbilical vein endothelial cells (HUVEC) subjected to Na,K-ATPase inhibition by ouabain or K^+^-free medium. According to our data, microRNAs, transcription factors, and proteins involved in immune response and inflammation might be considered as key components of Na^+^_i_/K^+^_i_-mediated excitation-transcription coupling. Special attention was focused on the *FOS* gene and the possible mechanism of transcription regulation via G-quadruplexes, non-canonical secondary structures of nucleic acids, whose stability depends on [Na^+^]_i_/[K^+^]_i_-ratio. Verification of the [Na^+^]_i_/[K^+^]_i_-sensitive transcription regulation mechanism should be continued in forthcoming studies.

## 1. Introduction

Gene expression is controlled by various stimuli in order to coordinate cellular metabolism and achieve cell-specific responses that ensure cell adaptation to the changed environment [1]. New technologies and approaches, developed over the past 25 years, have led to a more detailed understanding of the mechanisms, which provide different patterns of gene expression, evoked by the multiple stimuli in different types of cells [2]. The transcriptional response is usually very complex and often affects the activity of many hundreds of genes (the so-called transcription program). The response to environmental stimuli typically involves activation of immediate early response genes (the so-called “first wave” of expression) which encode transcription factors that affect activity of other genes (“second wave”). However, it is still difficult to discern the effect of a single stimulus on gene expression, because when applied to a cell the stimulus usually triggers several different signaling pathways that produce multiple messengers, which also induce gene expression.

The electrochemical gradient of Na^+^ and K^+^, existing across the plasma membrane in all animal cells, is maintained by Na,K-ATPase (Na-pump). This gradient is used by animal cells to support numerous physiological processes: generation of action membrane potentials (spikes), cell volume regulation, and secondary transport of ions (calcium, proton, phosphate, chloride) and nutrients (amino acids, nucleotides, sugars) [3]. An imbalance of intracellular Na^+^_i_ and K^+^_i_ concentrations, in response to different stimuli (e.g., hypoxia, inflammation, osmolality changes of extracellular fluids), can affect transcription and translation processes, thus contributing to the development of various diseases. Indeed, it has recently been demonstrated that an increase in [Na^+^]_i_/[K^+^]_i_-ratio in various types of cells results in differential expression of immediate early response genes, including *FOS* [4], as well as cell-specific genes, in particular, Na,K-ATPase α1-, α2-, and α3-subunits [5], tumor growth factor-β, skeletal muscle actin, myosin light chains, atrial natriuretic factor [6], and mortalin [7]. It was initially suggested that the alteration of gene expression caused by an increase in [Na^+^]_i_/[K^+^]_i_-ratio is associated with a change in the intracellular Ca^2+^ concentration, which is mediated by Na^+^/Ca^2+^ exchanger [8] and/or voltage dependent Ca^2+^-channels [9]. However, we have recently shown that ouabain, Na,K-ATPase inhibitor, triggers similar gene expression alterations in the presence of Ca^2+^ channel blockers and extra- and intracellular Ca^2+^ chelators in rat vascular smooth muscle cells (RVSM), human adenocarcinoma cell line (HeLa), and human umbilical vein endothelial cells (HUVEC) [10].

The data suggest that an increase in [Na^+^]_i_/[K^+^]_i_-ratio affects gene transcription through Ca^2+^_i_-independent mechanism(s), and that alteration of this parameter may be considered as a trigger of gene expression. Dissipation of the Na^+^_i_/K^+^_i_ gradient in endothelial cells can lead to their dysfunction, and thus contribute to the development of cardiovascular diseases. The object of this study was human umbilical vein endothelial cells. These cells are a convenient model for the investigations of pathophysiological aspects that contribute to the development of cardiovascular diseases [11]. Using Affymetrix technology, we tried to identify the first gene targets (upstream genes), whose expression change was induced by the elevation of [Na^+^]_i_/[K^+^]_i_-ratio. We performed a comparative analysis of time-dependent modulation of the [Na^+^]_i_/[K^+^]_i_-ratio and transcriptomic changes in HUVEC triggered by ouabain and K^+^-free medium, i.e., two independent approaches to Na,K-ATPase inhibition, with a final goal of identifying intermediates of the upstream signaling pathway. It is known that ouabain not only inhibits Na,K-ATPase, but, being bound to the pump molecule, also triggers some signaling pathways through protein–protein interactions [12]. This signaling affects gene expression too. We considered the differentially expressed genes common to both stimuli as Na^+^_i_/K^+^_i_ sensitive genes. Our results revealed upstream [Na^+^]_i_/[K^+^]_i_ sensitive gene targets, and allowed us to suggest how they might be regulated by the change in [Na^+^]_i_/[K^+^]_i_-ratio.

## 2. Results

### 2.1. Intracellular Na^+^ and K^+^ Content

In order to identify Na^+^_i_/K^+^_i_ sensitive genes, we investigated the kinetics of changes in their expression in endothelial cells in response to an increase in the intracellular [Na^+^]_i_/[K^+^]_i_-ratio. For this purpose, HUVEC were incubated in the presence of 3 μM ouabain or in K^+^-free medium for 0.5, 1, 2, and 6 h. Both stimuli provide inhibition of Na,K-ATPase and lead to the dissipation of the transmembrane gradient of monovalent cations.

Figure 1 shows that the dissipation of this gradient develops over time. In both cases (incubation with 3 μM ouabain and K^+^-free medium) the increase in the intracellular Na^+^_i_ content was approximately the same. In 6 h of Na,K-ATPase inhibition, this parameter increased five-fold relative to the control. Significant differences in the action of both stimuli were observed in the case of changes in the intracellular K^+^_i_ content. Thus, a drastic K^+^_i_ leakage occurred in the K^+^-free medium in 0.5 h (two-fold); this parameter reached its maximum value (20-fold) in 2 h. In the case of addition of 3 μM ouabain, a dramatic change in the intracellular K^+^_i_ content was observed in 2 h (three-fold). Both stimuli provided maximum inhibition of Na,K-ATPase in 6 h (Appendix A). This significant difference in the dissipation of the monovalent cations’ gradient can be explained by the fact that in the presence of ouabain only Na,K-ATPase is inhibited, while the Na,K,Cl-cotransporter, which provides about 20% of K^+^ transport into the cell, continues to work. Under conditions where cells are incubated in a K^+^-free medium, both systems providing intracellular K^+^ transport cease to function. At the same time, an increase in the intracellular Na^+^_i_ content leads to a change in the intracellular [Ca^2+^]_i_ concentration, which mediates the activation of small and intermediate conductance Ca^2+^-sensitive K^+^-channels, providing K^+^_i_ leakage [13].

### 2.2. Transcriptomic Changes Triggered by [Na^+^]_i_/[K^+^]_i_-Ratio Augmentation

In the current experiments, we examined the gene expression profile of HUVEC after 0.5, 1, 2, and 6 h of exposure to 3 µM ouabain and K^+^-free medium. The transcriptomic data obtained in three independent experiments were normalized, and then analyzed by principal component analysis (PCA), as described elsewhere [10]. This approach clearly demonstrated the dissimilarity between different experimental samples (Figure 2). Table 1 shows that 0.5 and 1 h incubation of HUVEC with ouabain resulted in the appearance of 30 and 16 genes, respectively, whose expression was changed by more than 20%, with *p* < 0.05. The number of differentially expressed genes was sharply increased in 2 and 6 h after ouabain addition (111 and 3415, respectively). With time-dependent inhibition of Na,K-ATPase in K^+^-free medium, the number of these genes was 64, 219, 507, and 3628, respectively. It can be noted that the number of genes that change their expression, as well as their fold of change, depend on [Na^+^]_i_/[K^+^]_i_-ratio, as we showed earlier [14,15].

Despite the fact that both stimuli (3 μM ouabain and K^+^-free medium) lead to the inhibition of Na,K-ATPase in HUVEC, we found some differences in their transcriptomic profiles. Figure 3 shows the distribution of common differentially expressed genes identified in cells exposed to 3 μM ouabain or subjected to K^+^-free medium vs. control. It is clear that with inhibition of Na,K-ATPase in 0.5 and 1 h, the number of common differentially expressed genes (DEGs) was eight. This value augmented with the increase of inhibition time and amounted to 53 and 2401 DEGs in 2 and 6 h, respectively. Importantly, the transcription changes evoked by ouabain and K^+^-free medium demonstrated remarkable positive correlation (Figure 4).

Thus, it can be assumed that the effect of ouabain on HUVEC leads to a change in gene transcription, not only mediated by augmentation of [Na^+^]_i_/[K^+^]_i_-ratio as a result of Na,K-ATPase inhibition. Nevertheless, an increase in [Na^+^]_i_/[K^+^]_i_-ratio is a sufficient condition for the transcriptomic changes. For the identification of upstream sensors involved in overall transcriptomic changes, we subdivided genes on three time-dependent categories. Early response genes (*ERG*) were defined as genes whose expression changed in 0.5 or 1 h of Na,K-ATPase inhibition by more than 1.2-fold. Intermediate response genes (*IRG*) were defined as genes whose expression changed in 2 h by more than 1.5-fold, with the exception of *ERG*. The rest of the differentially expressed genes detected in 6 h of Na,K-ATPase inhibition were defined as late response genes (*LRG*).

### 2.3. Analysis of Early Response Transcriptomic Changes Triggered by [Na^+^]_i_/[K^+^]_i_-Ratio Augmentation

Table 2 lists common genes whose expression changed by more than 1.2-fold upon inhibition of Na,K-ATPase by 3 μM ouabain and K^+^-free medium in 0.5 and 1 h (early response genes, *ERG*). After 0.5 h of Na,K-ATPase inhibition, the expression of the two genes, *RNU6-447P* and *RNU6-747P*, was increased. These genes encode U6 small nuclear RNAs which are the key components of spliceosome. Another six common genes showed reduced expression, among them *SNORD32B*, *SNORA20*, *MLF1*, *NR4A2, FOSB,* and *FOS*. *SNORD32B* and *SNORA20* encode the corresponding small nucleolar RNAs that are required for the maturation of other RNA molecules [16]. *MLF1*, *NR4A2*, *FOSB,* and *FOS* encode transcription factors. It should be noted that among these genes there are those that are associated with the regulation of cell passage through cell cycle. The product of *MLF-1* inhibits cell cycle passage via a p53-dependent mechanism [17]. In addition, some genes (*MLF1*, *SNORA20*) are associated with carcinogenesis, and are considered as tumor markers [18,19,20].

The number of differentially expressed genes increases as [Na^+^]_i_/[K^+^]_i_-ratio augments (Table 1). In addition, the set of common genes whose expression was triggered by both stimuli was changed. Thus, after 1 h of Na,K-ATPase inhibition, there was a statistically significant increase in *EGR1*, *FOS*, *LSMEM1*, *SNORA67*, *SNORA21*, *PPP1R10*, and *WDR47* expression, and a decrease in *MIR27B* expression. The products of the majority of these genes provide transcription regulation: *EGR1* and *FOS* encode the corresponding transcription factors, *SNORA67* and *SNORA21* small nucleolar RNAs, and *MIR27B* encodes microRNAs that are involved in the regulation of gene expression at the post-transcriptional level [21]. Thus, it can be assumed that at the early stages of dissipation of the Na^+^/K^+^ transmembrane gradient, the “rearrangement” of the transcription factors system takes place.

### 2.4. Analysis of Intermediate and Late Response Transcriptomic Changes Triggered by [Na^+^]_i_/[K^+^]_i_-Ratio Augmentation

After 2-h suppression of Na,K-ATPase by 3 μM ouabain or K^+^-free medium, a further increase in [Na^+^]_i_/[K^+^]_i_-ratio occurred, which was accompanied by an increase in the number of differentially expressed genes (Table 1). Among 53 common genes whose expression was evoked by both stimuli (Appendix A), we found genes whose products are involved in mRNA splicing and transcriptional regulation (e.g., *RNU-1263P*, *SNORD13E*, *TSC22D2*, *SNAPC1*, *HOXA9*), in the formation of pro-inflammatory response (e.g., *Il1**𝛼*, *CXCL8*, *Il1RL1*, *KITLG*), carcinogenesis (e.g., *SPRY4*, *CD274*), and intracellular signaling (e.g., *OR7E115P*, *GEM*, *GPR75*, *RND3*, *KITLG*). In addition, among them there were genes which encode molecules of the K^+^-channel, providing K^+^ entry into cells (*KCNJ2*), and Ca^2+^-dependent mitochondrial transporter of metabolites, nucleotides, and coenzymes (*SLC25A5*). Among the most upregulated genes we found *RN7SL600P*, *RN7SL473P,* and *RN7SL849P*, which are pseudogenes. Their biological functions are multiple, and often not detectable. However, it was shown that pseudogenes can affect chromatin and genomic structural organization, and are involved in the development of human diseases [22]. KEGG analysis showed that intermediate response transcriptomic changes in HUVEC generate a pro-inflammatory response (Appendix A).

A more significant dissipation of the transmembrane gradient of monovalent cations (after 6 h) led to a sharp increase in the number of genes whose expression was altered by more than 1.2-fold. Annotation of differentially expressed late response genes on the KEGG database demonstrated signaling pathways which respond for cell death activation (Appendix A). Thus, we could identify a general transcriptomic changes trend, depending on [Na^+^]_i_/[K^+^]_i_-ratio. Intermediate response genes are involved in the formation of a pro-inflammatory response and, possibly, in the reorganization of the chromatin structure. The late response genes set is characterized by an increase in the expression of proapoptotic factors, in contrast to the early response genes, among which there are those which encode proteins controlling cell division and also preventing apoptosis. Table 3 lists the top 10 intermediate and late response genes.

### 2.5. FOS Is a Conceivable Upstream Na^+^_i_/K^+^_i_-Sensitive Gene

The main question that arises during the study of Na^+^_i_/K^+^_i_ sensitive genes is: how do these cations regulate gene expression or, in other words, through which regulatory elements of DNA can Na^+^_i_/K^+^_i_-dependent transcriptomic changes occur? We focused our attention on the so-called G-quadruplexes, non-canonical secondary structures formed by guanine-rich regions of nucleic acids. It is known that G-quadruplexes are widespread in the genomes of prokaryotes and eukaryotes. It is assumed that they are involved in the regulation of transcription. It was also shown that their structure is stabilized by the monovalent cations, and that this depends on the ion radius in the following sequence K^+^ > Na^+^, NH_4_^+^ > > > Li^+^ [23]. Thus, it cannot be ruled out that an increase in [Na^+^]_i_/[K^+^]_i_-ratio leads to destabilization of these structures, and to subsequent transcriptomic changes. We examined early response genes (Table 2) for the possible presence of G-quadruplexes within their sequences. Using G4Catchall, an online tool for prediction of G-quadruplexes, we found that, hypothetically, these structures could be formed within *FOS, FOSB, LSMEM1, PPP1R10, WDR47, EGR1, NR4A2,* and *MLF1* genes, in 1–16 regions (Table 4 and Appendix A).

The *FOS* gene deserves special attention in this list. In the initial experiments, we observed that [Na^+^]_i_ elevation in HUVEC results in rapid accumulation of the RNAs encoding *FOS* [24]. In addition, according to our previous data, *FOS* expression in C2C12 myotubes occurs in a Ca^2+^_i_-independent manner [25]. These results, as well as early reports on Ca^2+^-independent *FOS* expression in RVSMC [4] and HeLa cells [26], allowed us to conclude that, along with canonical Ca^2+^_i_-mediated signaling, sustained elevation of [Na^+^]_i_/[K^+^]_i_-ratio affects gene transcription via unknown Ca^2+^_i_-independent mechanism(s). In this study, *FOS* expression decreased in cells in 0.5 h of exposure to 3 µM ouabain and K^+^-free medium. It began to increase in 1 h of Na,K-ATPase inhibition (the fold of change increased from −1.73 to + 1.96, and from −2.17 to 2.52, respectively). A further augment in [Na^+^]_i_/[K^+^]_i_-ratio in the presence of 3 µM ouabain elevated the expression of *FOS.* A similar, but more significant change in its expression was detected in the K^+^-free medium. Conversely, the time-dependent monovalent cations gradient dissipation in this case was accompanied by an increase in *FOS* expression for 1–2 h, and a subsequent decrease in its expression for 6 h (Table 5). *FOS* gene is an immediate early response gene, its protein products form one of the subfamilies of the AP-1 transcription factors family [27]. Transcription of a number of genes is carried out via *FOS*. Our study shows that *FOS* gene can be considered as an upstream Na^+^_i_/K^+^_i_ sensitive gene. Indeed, at the early stages of Na^+^_i_/K^+^_i_ gradient dissipation, a number of genes are controlled by *FOS* (Figure 5). Furthermore, their number and fold of change correlate with the [Na^+^]_i_/{K^+^]_i_-ratio. As mentioned above, *FOS* transcription decreased in 0.5 h of Na,K-ATPase inhibition in both cases. It should be noted that the total number of differentially expressed genes in ouabain-treated cells decreased in 1 h compared to 0.5 h of incubation (Table 1).

Figure 5 illustrates the network visualization of potential transcriptional regulation by *FOS* in HUVEC, triggered by Na,K-ATPase inhibition. As we can see, the 0.5 h HUVEC exposure to 3 µM ouabain downregulates *FOS*, *FOSB*, *MLF1*, and *NR4A2* (Table 2). In agreement with the transcriptional regulation network, these genes, including *FOS,* are regulated by *FOS* (Figure 5A). A similar situation is observed when cells are subjected to the K^+^-free medium (Table 2). However, in this case, the number of genes regulated by *FOS* is much larger (Figure 5B). Further augmentation of [Na^+^]_i_/[K^+^]_i_-ratio upregulates *FOS* expression, and leads to the multiplicity of early response genes set, regulated by *FOS* (Table 2, Figure 5B,C).

### 2.6. Activation of Signaling Pathways Involving Act, CREB, JNK, and ERK

In the current study, we also analyzed the activation of signaling pathways involving protein kinase B (Akt) and its target, the transcription factor CREB. It is known that activation of these signaling pathways leads to cell survival. In our experiments, we observed an increase in Akt phosphorylation at Ser473 with an increase in [Na^+^]_i_/[K^+^]_i_-ratio starting at 0.5 h of expose in the presence of ouabain and in K^+^-free medium; however, in 6 h the level of phosphorylation of this kinase significantly decreased (Figure 6A). Drastic activation of CREB, as in the case of Akt, occurred after 2 h of the Na^+^_i_/K^+^_i_ gradient dissipation, then the level of CREB phosphorylation also decreased (Figure 6B). Considering these data, it may be assumed that signaling pathways ensuring cell survival are activated at the early stages of dissipation of the gradient of Na^+^ and K^+^ ions.

It is well documented that the c-Fos/c-Jun complex becomes active under c-Fos and c-Jun phosphorylation by mitogen-activated protein kinases (MAPK), such as ERK and JNK [28,29]. MAPK/ERK and MAPK/JNK signaling pathways are essential for cellular responses to different stimuli [30]. Ouabain-induced activation of this signaling has been shown [31]. We observed that JNK increasing phosphorylation occurred in HUVEC treated by 3 µM ouabain for 0.5, 1, and 6 h. At the same time, the 2-h exposure of cells to ouabain did not have a similar effect (Figure 6C). On the contrary, in cells subjected to the K^+^-free medium, JNK activation was observed in 0.5, 1, and 2 h, but not in 6 h (Figure 6C). The most obvious activation of JNK in HUVEC was detected in 2 h in K^+^-free medium and in 6 h in the presence of ouabain. These data correlate with *FOS* gene expression changes (Table 5). Besides this, we may conclude that the observed effects were not primarily mediated by dissipation of the Na^+^_i_/K^+^_i_ gradient. ERK activation was marked in ouabain-treated cells in 0.5, 2, and 6 h (Figure 6D). Incubation of cells in K^+^-free medium led to increased ERK phosphorylation in 1, 2, and 6 h (Figure 6D). It should be noted that in 0.5 h of [Na^+^]_i_/[K^+^]_i_-ratio augmentation, caused by both stimuli, no significant ERK phosphorylation was detected (Figure 6D). Our recent findings showed that ERK, JNK, and CREB are implicated in Ca^2+^-independent signaling [25]. Summarizing these data, it may be assumed that *FOS* gene expression at the early stages of monovalent ions transmembrane gradient dissipation is regulated directly by [Na^+^]_i_/[K^+^]_i_-ratio changes. JNK and ERK kinases, most likely act indirectly, and display a cumulative effect in this case.

Thus, we can speculate that *FOS* is a potential Na^+^_i_/K^+^_I_ sensitive upstream gene, responsible for the subsequent transcriptomic changes in HUVEC.

## 3. Discussions

Numerous studies have shown that inhibition of Na,K-ATPase, resulting in the elevation of [Na^+^]_i_/[K^+^]_i_-ratio, leads to the alteration of gene expression in various types of cells. Among them were found RVSM, HUVEC, HeLa [10], cardiomyocytes [6], renal epithelial cells [32], hepatocytes [33], and neuronal cells [34]. Treatment of these cells by ouabain at concentrations that completely inhibit Na,K-ATPase induces expression of immediate early response genes, in particular, *FOS*. In this study we confirmed that 0.5–6 h incubation of HUVEC with ouabain or K^+^-free medium, i.e., two independent approaches to Na,K-ATPase suppression, affects gene expression. These transcriptomic changes are accompanied by the gain of Na^+^_i_ and loss of K^+^_i_, thus suggesting a key role of [Na^+^]_i_/[K^+^]_i_-mediated, rather than [Na^+^]_i_/[K^+^]_i_-independent, signaling.

According to the currently accepted paradigm, inhibition of Na,K-ATPase by ouabain and subsequent increase of [Na^+^]_i_/[K^+^]_i_-ratio results in the elevation of intracellular Ca^2+^ concentration, through the activation of Na^+^/Ca^2+^ exchanger [8], or conductivity of L-type voltage-gated Ca^2+^-channels [9]. Both types of these transporters were found in endothelial cells [35]. It is known that increase of Ca^2+^_i_, in turn, controls gene expression, mainly by CaMKI (II, III)/NFκB, calcineurine/NFAT, and CaMKII (IV)/CREB signaling pathways [36]. However, we previously found that Ca^2+^ depletion, triggered by addition of extra-and intracellular Ca^2+^ chelators, increased rather than decreased the number of ubiquitous and cell-type specific Na^+^_i_/K^+^_i_ sensitive genes in RVSM, HeLa, and HUVEC. These results as well as early reports of Ca^2+^-independent *c-Fos* expression in RVSMC [4] and HeLa cells [26] allowed us to conclude that, along with the canonical Ca^2+^_i_-mediated signaling, a sustained increase in [Na^+^]_i_/[K^+^]_i_-ratio affects gene transcription via unknown Ca^2+^_i_-independent mechanism(s).

Another method of gene expression activation by ouabain was disclosed by Xie and Askari for cardiomyocytes [12]. They demonstrated that after ouabain binding to Na,K-ATPase, the enzyme interacts with membrane protein-partners that, in turn, trigger signal cascades. The signaling pathways start from the interaction of Na,K-ATPase with Src kinase, phosphorylation of the epidermal growth factor receptor, and consequent activation of Ras and p42/44 mitogen-activated protein kinases.

Taking into account that ouabain itself can evoke signaling pathways that affect gene expression, we compared transcriptomic changes triggered by ouabain and K^+^-free medium. Thus, we were able to identify common genes, whose expression is caused by both stimuli and mainly depends on changes in [Na^+^]_i_/[K^+^]_i_-ratio (see Figure 3). This suggests that in both cases, there was basically a common (identical) trigger for the transcriptomic changes.

The main goal of this study was to reveal upstream Na^+^_i_/K^+^_i_ sensitive genes, namely, the genes that are the first targets of a trigger that induces the alteration of gene expression. We found that eight common, differentially expressed, genes were the same for both stimuli after 0.5 h of Na,K-ATPase inhibition by ouabain or in K^+^-free medium (Table 3). Furthermore, all of them have a similar expression pattern (activation or suppression). After 1 h of Na,K-ATPase inhibition, we also detected only eight common genes with a similar expression pattern for both stimuli. However, the set of common genes at this time point was altered compared to the set found after 0.5 h of Na,K-ATPase inhibition. The expression of seven of these genes was increased, while the expression of only one of them was decreased (Table 3). The sets and expression patterns of the eight revealed common genes, triggered after 0.5 and 1 h of Na,K-ATPase inhibition, were the same, and did not depend on whether inhibition was produced by ouabain or K^+^-free medium. Thus, we can conclude that these transcriptomic changes are triggered by the common (identical) starting stimulus, namely, the alteration of [Na^+^]_i_/[K^+^]_i_-ratio.

All genes whose expression changed after 0.5 and 1 h of Na,K-ATPase inhibition are “first wave” genes that are necessary for the alteration of the expression of other genes. Among their products we can see RNAs required for splicing, small nucleolar RNA, microRNAs which are involved in the regulation processes at the post-transcriptional level, and proteins which are transcriptional regulation factors. The data demonstrating that expression of some of them (e.g., *FOS* and *FOSB*) was downregulated after 0.5 h and then upregulated later, apparently demonstrates that during the first hour after Na,K-ATPase inhibition we observed the “reorganization” of the expression pattern of cells.

Special attention was focused on the *FOS* gene. This gene encodes one of the immediately expressed transcription factors, which induce a “second wave” of gene transcription. The *FOS* gene product (c-Fos protein) is a member of the leucine zipper proteins family, which, as homo- or heterodimers, bind to a specific region of double-stranded DNA, and which leads to varying effects on transcription. It is believed that c-Fos acts by combining short-term extracellular signals with long-term changes, through coordinated changes in the expression of target genes. Curiously, c-Fos is currently considered as a marker of neuronal activity [37]. It is known that neuronal activity is coupled with a significant change of [Na^+^]_i_/[K^+^]_i_-ratio. Activation of *FOS* expression was also observed in contracting C2C12 myotubes, where it also correlates with an increase in this parameter [38]. We have previously shown that *FOS* gene expression occurs via Ca^2+^-independent mechanism(s) [4,25]. It should be noted that this phenomenon was also noticed earlier. For example, Nakagawa et al., in the experiments with deletion within *FOS* gene promoter, have demonstrated that expression of this gene is mediated by ouabain via Na,K-ATPase inhibition, regardless of the cAMP/Ca^2+^ response element ability [39]. In another study, these authors noted that ouabain-induced *FOS* gene expression is due to intracellular monovalent cations perturbations [40].

The c-Fos proteins form heterodimers with Jun family proteins, and thus regulate transcription of many cellular genes through mitogen-activated protein kinases (MAPKs). It is known that Akt modulates *FOS* gene transcription [41]. It has also been shown that *FOS* gene promoters contain a serum response element (SRE) and a Ca^2+^/cAMP response element (CRE), which are activated by [Ca^2+^]_i_ increments in the cytoplasm and nucleus, respectively [42], as well as protein kinase B (Akt) and its target, the transcription factor CREB. Our data demonstrated CREB activation only in 2 h of Na,K-ATPase inhibition. Thus, we can assume that in the early stages of Na^+^_i_/K^+^_i_ gradient dissipation, the coordination of the transcription factors network is likely to occur, which, probably, is not mediated primarily by Ca^2+^ signaling.

It is well known that monovalent ions intracellular concentrations change temporarily in response to various stimuli. Orlov and Hamet have formulated a hypothesis which proposes monovalent ions as secondary messengers [43]. Data on monovalent cation intracellular sensors has been reviewed in recent publication [44]. In our opinion, G-quadruplexes are the most suitable candidate for this role [45]. These structures are abundant in the human genome and formed within guanine-rich regions of nucleic acids, DNA, and RNA. It was also shown that their structure is stabilized by monovalent cations [23]. Telomeres and the promoter region of genes are the main locations of DNA G-quadruplexes [46]. The formation, or the reorganization, of these structures within gene promoters can affect transcription processes sterically. We speculate that Na^+^_i_/K^+^_i_ imbalance mediates transcriptomic changes directly, via change of DNA conformation G-quadruplexes. Sen and Gilbert were the first to show that the ratio of Na^+^ and K^+^ ions is critical for the DNA G-quadruplexes’ formation, the so-called “sodium–potassium switch” [47]. These non-canonical structures occur in promoters of oncogenes, such as *MYC*, *MYB*, *FOS*, and *ABL* [48]. In addition, experimental data show that G-quadruplexes are more frequent in cancer cells in comparison with normal cells [49]. In this regard, these structures are currently considered as molecular anti-cancer targets. It is worth mentioning that viruses also have G-quadruplexes in the regulatory regions of the genome, which are the anti-viral targets [50]. These observations are consistent with the data demonstrated by the application of cardiotonic steroids (CTS) in anti-cancer [51,52,53] and anti-viral therapy [54,55,56,57]. It is well documented that different members of the CTS family have different affinities to the Na,K-ATPase and, as a consequence, different inhibitory properties [58,59]. It is likely that the various therapeutic effects of these compounds are due to these parameters.

## 4. Materials and Methods

### 4.1. Cell Culture

The human umbilical vein endothelial cells (HUVEC) were purchased from Lonza (Maryland, Walkersville, MD, USA) and passaged up to 6 times. Cells were cultured in complete endothelial cell growth medium-2 (EGM-2 BulletKit, CC3162, Lonza, Maryland, Walkersville, MD, USA) containing 10% fetal bovine serum (FBS), and maintained in a humidified atmosphere with 5% CO_2_/balance air at 37 °C. To establish quiescence, cells were incubated for 24 h in media in which concentration of FBS was reduced to 0.2%.

### 4.2. Intracellular Na^+^ and K^+^ Content

Intracellular content of exchangeable Na^+^ and K^+^ was measured as the steady-state distribution of extra- and intracellular ^86^Rb and ^22^Na, respectively. To establish isotope equilibrium, cells growing in 12-well plates were preincubated for 3 h in control medium containing 0.5 µCi/mL ^86^RbCl or 3 µCi/mL ^22^NaCl, and then with or without 3 µM ouabain, and in K^+^-free medium for 0.5, 1, 2, and 6 h. The cells were transferred onto ice, washed 4 times with 2 mL of ice-cold medium W containing 100 mM MgCl_2_ and 10 mM HEPES-tris buffer (pH 7.4). The medium was aspirated and cells were lysed with 1% SDS and 4 mM EDTA solution. Radioactivity of incubation media and cell lysates was quantified, and intracellular cation content was calculated as A/a × m, where A was the radioactivity of the samples (cpm), a was the specific radioactivity of ^86^Rb (K^+^ congener) and ^22^Na in the medium (cpm/nmol), and m was the protein content. For more details, see [10].

### 4.3. RNA Isolation

Total RNA was extracted from cells grown in 6-well plates, using TRIzol^®^ reagent (Invitrogen, California, Carlsbad, CA, USA) and purified with a RNeasy^®^ MinElute cleanup kit (Qiagen, California, Valencia, CA, USA), following the manufacturers’ protocols. Only the RNA samples that had a more than 7.0 RNA integrity number (RIN) and no detectable genomic DNA contamination were used for the subsequent gene array analyses. RNA quality was assessed by 2100 Bioanalyzer (Agilent Technologies, Palo Alto, CA, USA). Microarray experiments were performed with GeneChip^®^ Human Gene 1.0 ST array (which detects 28,869 gene products). In this case, each gene was represented by approximately 26 probes along the entire length of the transcript (Affymetrix, California, Santa Clara, CA, USA). 100 ng of total RNA for each sample was processed with Ambion^®^ WT Expression Kit (Invitrogen). This kit uses a reverse transcription priming method that specifically primes non-ribosomal RNA, including both poly(A) and non-poly(A) mRNA, and generates sense-strand cDNA as the final product. 5.5 µg of the single-stranded cDNA was fragmented and labeled using a Affymetrix GeneChip^®^ WT Terminal Labeling Kit, and 2.0 µg of the resulting cDNA was hybridized on the chip.

### 4.4. Gene Chip Expression Analysis

The whole hybridization procedure was conducted with the Affymetrix GeneChip^®^ system, according to the protocol recommended by the manufacturer. The hybridization results were evaluated with Affymetrix GeneChip^®^ Command Console software (AGCC, version 0.0.0.676). Quality of the chips was determined using Affymetrix Expression Console. Data analysis was performed within Partek Genomics Suite (Partek, Missouri, St. Louis, MO, USA). The data were initially normalized by the robust multichip average (RMA) algorithm, which uses background adjustment, quantile normalization, and summarization. Then, normalized data were analyzed by principal component analysis (PCA) [60] to identify patterns in the dataset, and highlight similarities and differences among the samples. Major sources of variability identified within the dataset by PCA were used as grouping variabilities for analysis of variance (ANOVA), with *n* = 3 for each group of samples. The ensuing data were filtered to identify transcripts with statistically significant variation of expression among the groups that are modulated by at least 20%, with multiple testing correction by the false discovery rate (*p*-VALUE).

### 4.5. GO and KEGG Pathways Analysis

Active subnetwork search and enrichment analysis was provided by the pathfindR package in R [61]. Biogrid and KEGG sets were used for the identification of protein–protein interaction network (PIN) and necessary gene sets to obtain for enrichment analysis, respectively. A greedy search (GR) algorithm was used to perform an active subnetwork search. Network of the set of co-regulated genes was predicted by using iRegulon in Cytoscope software [62]. The Ggplot2 package was used for data visualization [63].

### 4.6. G-Quadruplex-Forming Region Analysis

The search for quadruplexes within the DNA sequences of early response genes (0.5 and 1 h) was performed using the G4Catchall online tool [64], NCBI Nucleotide, and Gene databases. Values with a score greater than 0 were considered reliable.

### 4.7. Western Blot Analysis

The harvested lysates of cells were prepared by solubilization in RIPA buffer, containing cocktails of proteases and phosphatases inhibitors (Thermo Fisher Scientific, Massachusetts, Waltham, MA, USA). Protein concentrations were determined using the Lowry protein assay [65]. SDS-PAGE was performed in accordance with the Laemmli method [66], with 6% stacking gel and 10% running gel. Proteins were transferred from gel to nitrocellulose membrane (BioRad, California, Hercules, CA, USA), followed by blocking with 5% fat-free milk (Valio, Helsinki, Finland) in PBST 1X (PBS supplemented with Tween 20 0.1%) for 1 h at room temperature. Akt, CREB, and their phosphorylated forms were detected by overnight incubation with the corresponding antibodies (#9272S, #4060S, #4820S, #9198S, #9252S, #4668S, #4695S #4377S, Cell Signaling, Massachusetts, Danvers, MA, USA) at 4 °C in 5% milk powder in PBST. These samples were then incubated with HRP-conjugated secondary antibodies monoclonal anti-rabbit IgG (a1949, Sigma-Aldrich, Missouri, St. Louis, MO, USA) for 1 h at room temperature in 5% milk powder in PBST. GAPDH (#2118S, Cell Signaling, Massachusetts, Danvers, MA, USA) was chosen as the house-keeping protein. Antigen-antibody complexes were visualized using an ECL kit and ChemiDoc XRS+ Molecular Imager (BioRad, California, Hercules, CA, USA).

### 4.8. Chemicals

^22^NaCl and ^86^RbCl were obtained from PerkinElmer (Massachusetts, Waltham, MA, USA) and Isotope (St-Petersburg, Russia). The remaining chemicals were supplied by Gibco BRL (Missouri, Gaithersburg, MO, USA), Calbiochem (California, La Jolla, CA, USA), Sigma-Aldrich (Missouri, St. Louis, MO, USA), and Anachemia Canada Inc. (Quebec, Montreal, QC, Canada).

## 5. Conclusions

Our study shows that changes in the expression of major early, and intermediate, response genes are mediated almost exclusively by an increase in [Na^+^]_i_/[K^+^]_i_-ratio, rather than Na^+^_i_/K^+^_i_-independent signaling. Among Na^+^_i_/K^+^_i_-sensitive genes, there are those that deserve special attention. These genes, encoding microRNAs, transcription factors, and proteins, are involved in the formation of the immune and inflammatory response. They can be key in the system of excitation-transcription coupling, due to the dissipation of the gradient of monovalent cations. Na^+^_i_/K^+^_i_-dependent changes in gene transcription can be carried out with the participation of G-quadruplexes, non-canonical secondary structures of nucleic acids formed by guanine-rich regions of these acids. It was shown that their structure is stabilized by the monovalent cations. Thus, it cannot be ruled out that an increase in [Na^+^]_i_/[K^+^]_i_-ratio leads to destabilization of these structures and a subsequent change in gene transcription. However, this assumption needs to be experimentally confirmed.

## Figures and Tables

**Figure 1 ijms-21-07992-f001:**
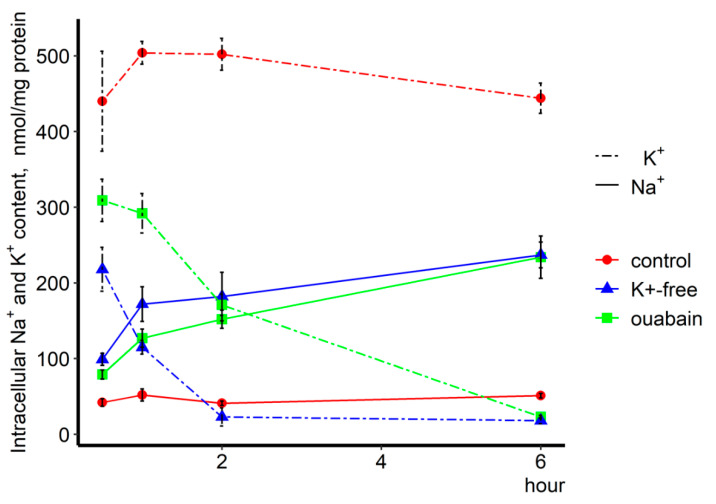
Kinetics of modulation of intracellular Na^+^ and K^+^ content by Na,K-ATPase inhibition in human umbilical vein endothelial cells. Cells were exposed to 3 µM ouabain and K^+^-free medium for 0.5, 1, 2, and 6 h. The Na^+^_i_ and K^+^_i_ content in cells for each time point in the absence of treatment was taken as 100%. Means ± S.E. from three experiments performed in quadruplicate are shown.

**Figure 2 ijms-21-07992-f002:**
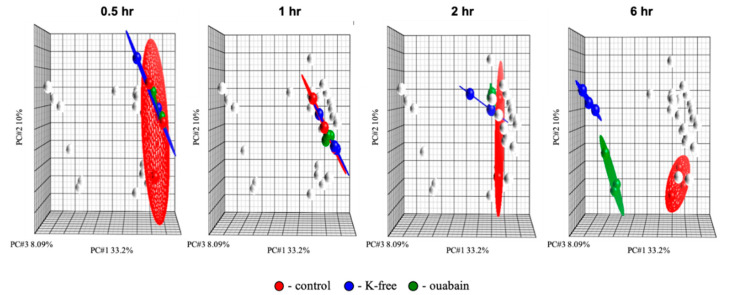
Principal component analysis (PCA) of the transcriptomic changes in human umbilical vein endothelial cells (HUVEC) triggered by Na,K-ATPase inhibition by 3 µM ouabain and incubation in K^+^-free medium for 0.5, 1, 2, and 6 h. Ellipsoids highlight portioning of samples based on the type of treatment. The principal components in the three-dimensional graphs (PC #1, PC #2, and PC #3) represent the variability of gene expression level within the datasets. Each point on the PCA represents the gene expression profile of an individual sample. Samples that are near each other in the resulting three-dimensional plot have a similar transcriptome, while those that are further apart have dissimilar transcriptional profiles. White dots, as well as green, red and blue, represent overall statistically significant transcriptomic changes. In turn, green, red and blue dots show the differentially expressed genes sets specific to each stimulus which change at different time points. All experiments were repeated three times.

**Figure 3 ijms-21-07992-f003:**
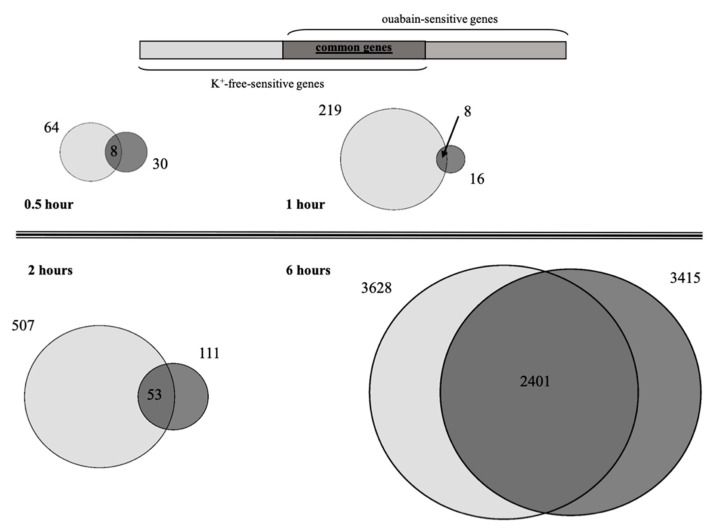
Venn diagram representing the distribution of common differentially expressed genes identified in HUVEC exposed to 3 μM ouabain, or subjected to K^+^-free medium, vs. control samples in 0.5, 1, 2, and 6 h. The number of common differentially expressed genes are shown within overlapping areas. The total number of upregulated and downregulated genes identified in ouabain- or K^+^-free medium-treated cells is shown in Table 1.

**Figure 4 ijms-21-07992-f004:**
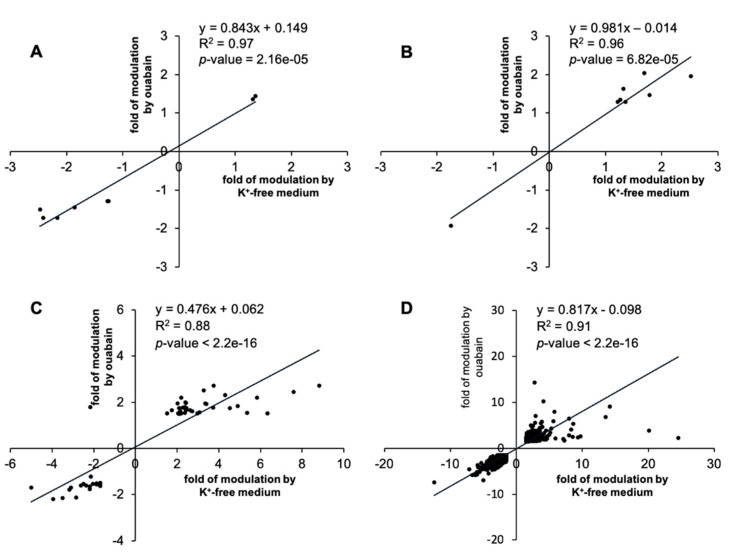
Correlation analysis of genes whose expression was changed by Na,K-ATPase inhibition in the presence of 3 μM ouabain or K^+^-free medium in 0.5 (**A**), 1 (**B**), 2 (**C**), and 6 h (**D**) by more than 1.2-fold with *p* < 0.05. The total number of genes subjected to analysis is shown in Figure 2.

**Figure 5 ijms-21-07992-f005:**
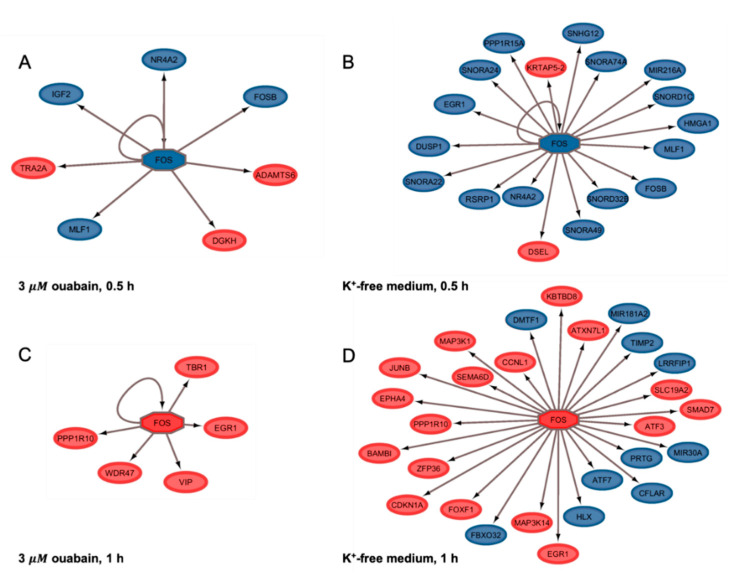
Potential transcriptional regulation in HUVEC by FOS, triggered by Na,K-ATPase inhibition by 3 µM ouabain (**A**,**C**) or K^+^-free medium (**B**,**D**) at the early stages of Na^+^_i_/K^+^_i_ gradient dissipation. Upregulated and downregulated genes are shown in red and blue, respectively.

**Figure 6 ijms-21-07992-f006:**
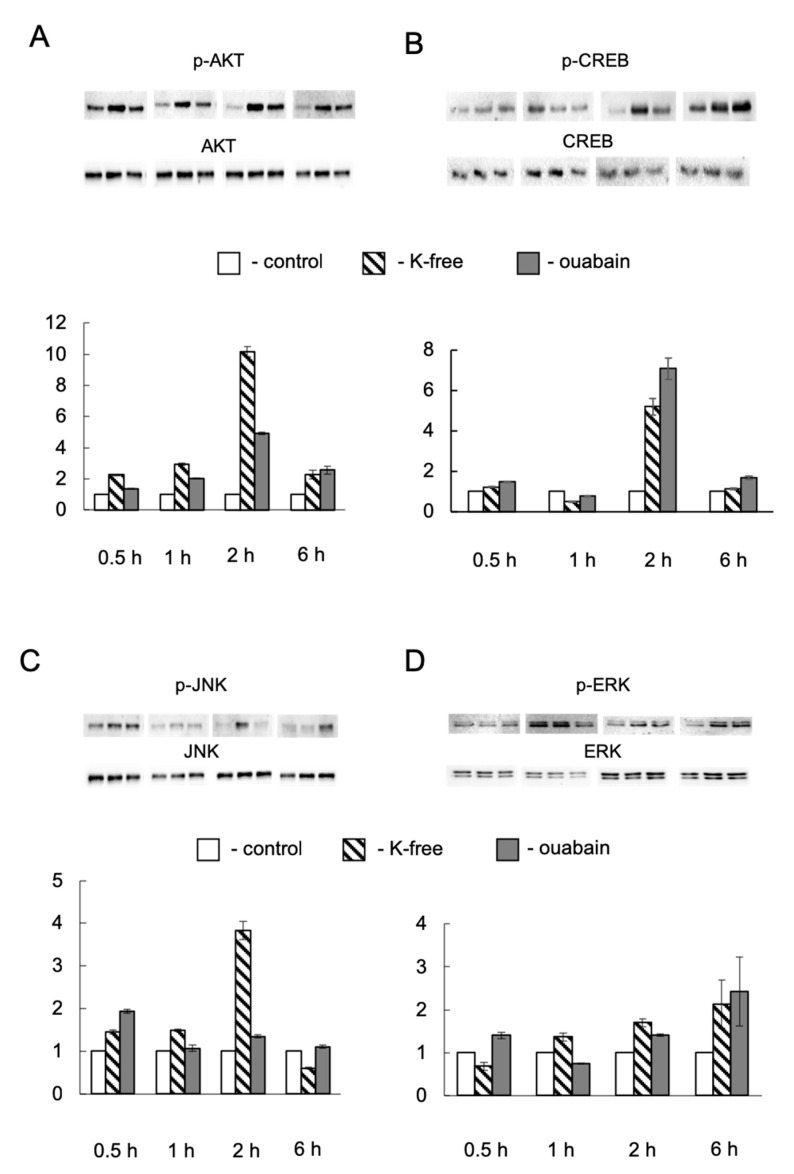
Representative western blots and relative content of phosphorylated Akt (**A**), CREB (**B**), JNK (**C**), and Erk (**D**) in control HUVEC and after 0.5-, 1-, 2-, and 6-h exposure to 3 µM ouabain or K^+^-free medium. The content of proteins in control cells was taken as 1.0. Data obtained in three independent experiments are reported as means ± S.E.

**Table 1 ijms-21-07992-t001:** Time-dependent effect of Na,K-ATPase inhibition on genes expression in HUVEC.

	Ouabain				K^+^-Free Medium			
	0.5 h	1 h	2 h	6 h	0.5 h	1 h	2 h	6 h
Number of differentially expressed genes	30	16	111	3415	64	219	507	3628
Number of up-regulated genes	22	15	64	1231	6	133	280	1011
Maximal fold of activation	1.56	2.04	2.73	14.49	1.48	2.55	17.22	24.57
Number of down-regulated genes	8	1	44	2184	58	86	227	2617
Maximal fold of inhibition	−1.73	1.93	−2.20	−7.49	−2.47	−3.44	−4.99	−12.42

Genes whose expression was altered by more than 1.2-fold with *p* < 0.05 for 0.5 h and 1 h, and more than 1.5-fold with *p* < 0.05 for 2 h and 6 h were subjected to analysis.

**Table 2 ijms-21-07992-t002:** Common differentially expressed genes triggered by Na,K-ATPase inhibition by 3 μM ouabain or K^+^-free medium in 0.5 and 1 h.

Hours	Gene Symbol	3 μM Ouabain		K^+^-Free Medium	
		FC	*p*-Value	FC	*p*-Value
0.5	RNU6–447P	1.44	1.23 × 10^3^	1.36	5.26 × 10^3^
	RNU6–747P	1.36	1.48 × 10^3^	1.32	5.26 × 10^3^
	SNORD32B	−1.29	1.21 × 10^2^	−1.26	2.43 × 10^2^
	MLF1	−1.29	6.26 × 10^3^	−1.27	9.23 × 10^3^
	SNORA20	−1.45	476 × 10^2^	−1.86	6.49 × 10^4^
	NR4A2	−1.51	6.26 × 10^3^	−2.47	7.12 × 10^8^
	FOSB	−1.73	3.11 × 10^3^	−2.42	5.16 × 10^6^
	FOS	−1.73	3.71 × 10^2^	−2.17	2.72 × 10^3^
1	EGR1	2.04	1.90 × 10^3^	1.69	8.07 × 10^3^
	FOS	1.96	1.19 × 10^3^	2.52	3.65 × 10^5^
	LSMEM1	1.63	1.41 × 10^3^	1.32	2.92 × 10^2^
	SNORA67	1.47	5.42 × 10^3^	1.78	3.33 × 10^6^
	SNORA21	1.35	1.06 × 10^2^	1.26	2.23 × 10^2^
	PPP1R10	1.29	1.90 × 10^3^	1.21	6.48 × 10^3^
	WDR47	1.29	3.80 × 10^2^	1.36	7.36 × 10^4^
	MIR27B	−1.93	1.49 × 10^3^	−1.75	9.21 × 10^4^

Genes whose expression was altered by more than 1.2-fold with *p* < 0.05 were subjected to analysis.

**Table 3 ijms-21-07992-t003:** Top 10 common differentially expressed transcripts triggered by Na,K-ATPase inhibition by 3 µM ouabain or K^+^-free medium in 2 and 6 h.

Hours	Gene Symbol	3 µM Ouabain		K^+^-Free Medium	
		FC	*p*-Value	FC	*p*-Value
2	RN7SL600P	2.71	8.26 × 10^6^	8.81	9.24 × 10^13^
	RN7SL473P	2.45	1.03 × 10^7^	7.58	1.26 × 10^15^
	RN7SL849P	1.53	6.66 × 10^3^	6.33	6.76 × 10^14^
	IL1A	2.21	8.31 × 10^9^	5.83	1.18 × 10^16^
	KITLG	1.53	5.75 × 10^5^	5.36	1.11 × 10^16^
	FLJ35409	−2.13	2.02 × 10^3^	−2.85	5.98 × 10^6^
	AC002350.1	−1.78	3.43 × 10^3^	−3.16	4.24 × 10^8^
	LOC100130713	−2.14	9.38 × 10^7^	−3.49	1.12 × 10^11^
	SNORD52	−2.20	1.72 × 10^6^	−3.96	7.64 × 10^12^
	DEPP1	−1.69	8.84 × 10^6^	−4.99	9.57 × 10^16^
6	FLJ43390	14.33	2.91 × 10^17^	2.74	3.40 × 10^9^
	TFPI2	9.07	4.76 × 10^18^	14.21	8.28 × 10^20^
	HIVEP2	6.83	9.42 × 10^19^	13.52	1.80 × 10^21^
	PLA2G4C	6.48	1.63 × 10^18^	8.03	1.19 × 10^19^
	TRAF1	3.85	2.23 × 10^15^	20.12	3.62 × 10^22^
	AGGF1	−5.68	7.59 × 10^21^	−5.94	2.60 × 10^21^
	GCNT1	−5.78	2.51 × 10^16^	−6.16	8.77 × 10^17^
	NMI	−5.86	5.04 × 10^19^	−6.64	8.28 × 10^20^
	SAMD9L	−6.92	1.61 × 10^19^	−5.01	3.19 × 10^18^
	GIMAP2	−7.38	2.77 × 10^19^	−12.42	2.03 × 10^21^

Genes whose expression was altered by more than 1.5-fold with *p* < 0.05 were subjected to analysis.

**Table 4 ijms-21-07992-t004:** Prediction of G-quadruplexes within early response genes’ DNA.

Sequence Definition	NCBI Reference Sequence	Beginning of G4 Sequence (First Nucleotide = 0)	End of G4 Sequence	Length	G4-Forming Sequence	**G4HScore**
FOS	NC_000014.9	341	368	27	ggggccgggggcttggggtcgcggagg	2
FOS	NC_000014.9	1405	1427	22	gggaatgtgggggctgggtggg	2.136
FOS	NC_000014.9	2755	2778	23	gtgagggggcagggaaggggagg	2.174
FOSB	NC_000019.10	2242	2265	23	ggggtgggggtggggtgttgtgg	2.522
FOSB	NC_000019.10	3910	3930	20	gggaggtagagagggagggg	2
FOSB	NC_000019.10	5323	5348	25	ggggatgggtggggaggggggcggg	2.92
LSMEM1	NC_000007.14	5835	5858	23	gggtgtggtggagggggaggggg	2.522
LSMEM1	NC_000007.14	6220	6244	24	gggttggagaaagggggtgggggg	2.417
LSMEM1	NC_000007.14	9144	9167	23	gggttgggactgggagggagggg	2.217
PPP1R10	NC_000006.12	3449	3473	24	ggggtgtggggggggggttgcagg	2.542
PPP1R10	NC_000006.12	4975	4995	20	ggagcagttgggtggggggg	2
WDR47	NC_000001.11	61981	62015	34	gggtggggtgggaaggggtagggatggatggtttaggg	2
MLF1	NC_000003.12	2170	2225	55	ggggcggggcggggggaggggggcgggaggagggaggagggaggagggcggcgggggggggggggcggcgggggggggggggtgtgtgtg	2.633
MLF1	NC_000003.12	27114	27155	41	ggtggggggacaggagggaggtgtgggggttgggggtaagg	2.171

G-quadruplex-forming regions with G4HScore ≥ 2 are listed.

**Table 5 ijms-21-07992-t005:** Kinetics of *FOS* gene expression triggered by Na,K-ATPase inhibition.

Hours	3 µM Ouabain		K^+^-Free Medium	
	FC	*p*-Value	FC	*p*-Value
0.5	−1.73	3.71 × 10^2^	−2.17	2.72 × 10^3^
1	1.96	1.19 × 10^2^	2.52	3.65 × 10^5^
2	2.47	1.80 × 10^5^	18.07	1.00 × 10^15^
6	8.14	8.44 × 10^14^	5.68	2.64 × 10^12^

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
