# Peer review of "Transcriptomic Changes in Endothelial Cells Triggered by Na,K-ATPase Inhibition: A Search for Upstream Na+i/K+i Sensitive Genes"

_ijms, 2020, doi:10.3390/ijms21217992_

Round 1

Reviewer 1 Report

Introduction. The rationale for this study should be added.

Results. Figure 1: The information on this figure is not easy to understand. Please, use the different colors for different conditions and more detailed explanation in the Legend.

Figure 2: It is not clear what group the grey dots belong to? They do not overlap with the groups indicated by the color dots standing for control, K-free, and ouabain-treated. In addition, the PCA are the same on four panels, although the panels presented the results for different time points.

Figure 3: Is the gene expression given for ouabain-treated cells vs. control and K+-free-treated cells vs. control? This should be explained in the Legend and in the Methods or the Results.

Figure 5: it might be better to use the colors: green – for the downregulated genes, and red – for the upregulated genes. In the present form, the graph is not visualizing the findings.

Methods: please, explain why relatively high concentration (3 micromoles) of ouabain was used? This concentration is much higher than concentration of the endogenous cardiotonic steroids, thus, the present findings are related to the toxic effect of ouabain. Also, why HUVEC cells were used? This should be explained and discussed or added to the study limitation.

The Discussion is too long; please, shorten.

Author Response

Dear reviewer, the authors thank you for your careful reading of the manuscript and valuable comments. Сhanges in the text are highlighted in gray. Please see the attachment.

  1. Introduction. The rationale for this study was added (47-49; 62-65).
  2. Figure 1. Changes applied (153-155).

  3. Figure 2. Normalized data analyzed by principal component analysis (PCA) of the transcriptome was carried out on all genes under investigation to identify patterns in the dataset and highlight similarities and differences among the samples. Grey dots, as well as green, red and blue, represent overall statistically significant transcriptomic changes (thus, the PCA are the same on four panels). In turn, green, red and blue dots show the differentially expressed genes sets specific to each stimulus which change at different time points (173).

  4. Figure 3. Changes applied (192).

  5. Figure 5. Changes applied (323).

  6. Methods. We performed a comparative analysis of time-dependent modulation of [Na+]i/[K+]i-ratio and transcriptomic changes in HUVEC triggered by ouabain and K+-free medium, i.e. two independent approaches to Na,K-ATPase inhibition, with a final goal to identify intermediates of upstream signaling pathway (67-70). Since ouabain was used by us as a tool to change the gradient of monovalent cations, we used it in high doses (3 mkM). Dissipation of the Na+i/K+i gradient in endothelial cells can lead to their dysfunction and thus contribute to the development of cardiovascular diseases. The object of this study was human umbilical vein endothelial cells. These cells are a convenient model for the investigations of pathophysiological aspects that contribute to the development of cardiovascular disease (62-65).
  7. The discussion was shortened.

Reviewer 2 Report

It is a well written manuscript and well structured research project.

Author Response

Dear reviewer, the authors thank you for your careful reading of the manuscript.

Round 2

Reviewer 1 Report

The Authors addressed all my recommendations. I do not have additional comments.